# Modeling serological testing to inform relaxation of social distancing for COVID-19 control

Alicia N. M. Kraay [1,5] ✉, Kristin N. Nelson[1,5], Conan Y. Zhao[2,3], David Demory[2], Joshua S. Weitz [2,4,5] & Benjamin A. Lopman [1,5]

Serological testing remains a passive component of the public health response to the COVID-19 pandemic. Using a transmission model, we examine how serological testing could have enabled seropositive individuals to increase their relative levels of social interaction while offsetting transmission risks. We simulate widespread serological testing in New York City, South Florida, and Washington Puget Sound and assume seropositive individuals partially restore their social contacts. Compared to no intervention, our model suggests that widespread serological testing starting in late 2020 would have averted approximately 3300 deaths in New York City, 1400 deaths in South Florida and 11,000 deaths in Washington State by June 2021. In all sites, serological testing blunted subsequent waves of transmission. Findings demonstrate the potential benefit of widespread serological testing, had it been implemented in the pre-vaccine era, and remain relevant now amid the potential for emergence of new variants.

[1] Rollins School of Public Health, Emory University, Atlanta, GA, USA. [2] School of Biological Sciences, Georgia Institute of Technology, Atlanta, GA, USA. [3] Interdisciplinary Graduate Program in Quantitative Biosciences, Georgia Institute of Technology, Atlanta, GA, USA. [4] School of Physics, Georgia Institute of Technology, Atlanta, GA, USA. [5] These authors contributed equally: Alicia N. M. Kraay, Kristin N. Nelson, Joshua S. Weitz, Benjamin A. Lopman.
✉ email: amullis@emory.edu

SARS-CoV-2 emerged in China in late 2019 leading to the COVID-19 pandemic, with over 213 million detected cases and over 4.4 million deaths globally and approximately 38 million detected cases and 643,000 deaths reported in the U.S. as of August 24, 2021[1]. Unprecedented social distancing measures were enacted in early 2020 to reduce transmission and blunt the epidemic peak. In March 2020, U.S. states began to close schools, suspend public gatherings, and encourage employees to work from home if possible. By mid-April, 95% of the U.S.[2] and over 30% of the global population were under some form of shelter-in-place order[3]. Federal social distancing guidelines expired on April 30, 2020; throughout the summer, many state and local governments relaxed stay-at-home orders partially or completely[4].

Relaxing these social distancing policies resulted in increased community transmission, and case counts increased as states further relaxed restrictions on public gatherings, restaurant dining, and operation of businesses[5]. Behavioral change combined with the accelerated transmission in a largely immunologically naïve population resulted in a wave of cases and deaths in the late summer and early Fall 2020, a second, larger wave in the Fall, and then a third wave in the Winter of 2020. During late spring 2021, the widespread availability of SARS-CoV-2 vaccines in the United States coupled with higher levels of natural immunity allowed social distancing interventions to be relaxed further. Despite the widespread availability of vaccines, a fourth wave of cases in the US beginning in late Summer 2021 is due to multiple factors, including fatigue from adhering to strict social distancing measures and heterogeneous vaccine coverage. The rise of more transmissible variants of concern[6,7] as well as the possibility of variants that escape natural or vaccine-derived immunity[8] continues to require vigilance in the event that COVID-19 incidence increases again. Indeed, this fourth wave reinforces the need to evaluate other measures—including individualized policies based on disease or immune status—as part of integrative response campaigns[9].

In this paper, we explore how immune shielding could be used to further reduce population risk. A shielding strategy aims to identify and deploy recovered/vaccinated (and likely immune) individuals as focal points for sustaining less risky interactions. This strategy has the objective of sustaining interactions necessary for the functioning of essential services while reducing the risk of exposing individuals who remain susceptible to infection. As the basis for a shielding strategy, widespread serological testing programs have the potential to identify individuals or groups who are likely immune, allowing some individuals to return to activities while keeping deaths and hospital admissions at sufficiently low levels. In this strategy, individuals who test positive would preferentially replace susceptible individuals in close-contact interactions, such that more contacts are between susceptible and immune individuals rather than between susceptible and potentially infectious individuals[10]. Immune shielding may be particularly useful given the high incidence in focal regions resulting from incomplete vaccine coverage and partial levels of population immunity[11].

Serosurveys of SARS-CoV-2 in the U.S. vary in their estimates of seroprevalence but collectively suggest that infections far outnumber documented cases[12–16]. To the extent that antibodies serve as a correlate of immunity, serological testing may be used to identify protected individuals[17]. While our understanding of the immunological response to SARS-CoV-2 infection remains incomplete, the vast majority of infected individuals seroconvert[18], with detectable antibody levels persisting at least several months after infection for the majority of individuals[19]. SARS-CoV-2 reinfections have been documented but remain relatively rare (though emerging variants of concern have raised questions regarding breakthrough infection rates[20]). Together, these data suggest that recovered individuals have substantial protection against subsequent re-infection. Once identified, antibody test-positive individuals could return to pre-pandemic levels of social interactions and therefore dilute (via shielding) potentially risky interactions between susceptible and infectious individuals[10]—keeping in mind that variants of concern may require that other NPIs are still utilized (e.g., masking in indoor settings).

Such strategies, however, rely on correctly identifying immune individuals. There are currently more than 50 serological assays for the detection of SARS-CoV-2 antibodies that have been authorized for emergency use by the Food and Drug Administration[21]. The performance of these tests varies considerably[21–23]. For the purpose of informing safe social distancing policies, specificity rather than sensitivity is of primary concern. An imperfectly specific test will result in false positives, leading to individuals being incorrectly classified as immune. If used as a basis to relax social distancing measures, there is concern that this error could heighten the risk for individuals who test positive and lead to an increase in community transmission. For this reason, this paper evaluates the integration of serological testing into a COVID-19 transmission model to evaluate the level of serological testing needed to reduce expected fatalities while increasing the fraction of focal populations who can re-engage in socio-economic activities.

## Results

To evaluate the epidemiological consequences of using mass serological testing to inform the relaxation of social distancing measures in the pre-vaccine era, we modeled transmission dynamics and serological testing for SARS-CoV-2 using a deterministic, compartmental SEIR-like model (Fig. 1). Recovered, susceptible, latently infected, and asymptomatic persons test positive at rates that are functions of testing frequency, sensitivity (for recovered individuals), and specificity (for non-immune individuals). We model contacts at home, work, school, and other locations among three age groups: children and young adults (< 20 years), working adults (20–64 years), and elderly (65+ years). We used a Markov Chain Monte Carlo (MCMC) approach to fit the model to time series of deaths[24] and cross-sectional seroprevalence[16] data from three U.S. metropolitan areas with distinct COVID-19 epidemic trajectories: the New York City Metro Region, South Florida, and the Washington Puget Sound region, including changes in policy impacting social distancing behaviors, to evaluate an immunological shielding strategy per-region.

**Model fits to fatalities and serological data**. We explored the impacts of social distancing on epidemic outcomes in the absence of serological testing. To do so, we first used MCMC model-data integration to fit the model to reported deaths and seroprevalence point estimates from each of three metropolitan areas. Model fits reproduced reported death trends reasonably well through June 2020 and seroprevalence estimates early in the outbreak (see Fig. 2 for fits; Supplementary Figs. 1–10 for full model diagnostics). Of note, fits were poorer for New York City, which was not unexpected due to the unique severity of the initial pandemic wave there. Fits were moderately good for Washington Puget Sound and best for South Florida. We note that the probability of infection per contact had narrow credible intervals (CrI), indicating posterior confidence in the ability of the model to uniquely identify parameter sets consistent with key features of infection. Credible intervals were wider for the fraction of infections that were symptomatic and were widest for social distancing parameters, indicating limits of parameter identifiability. Nonetheless,

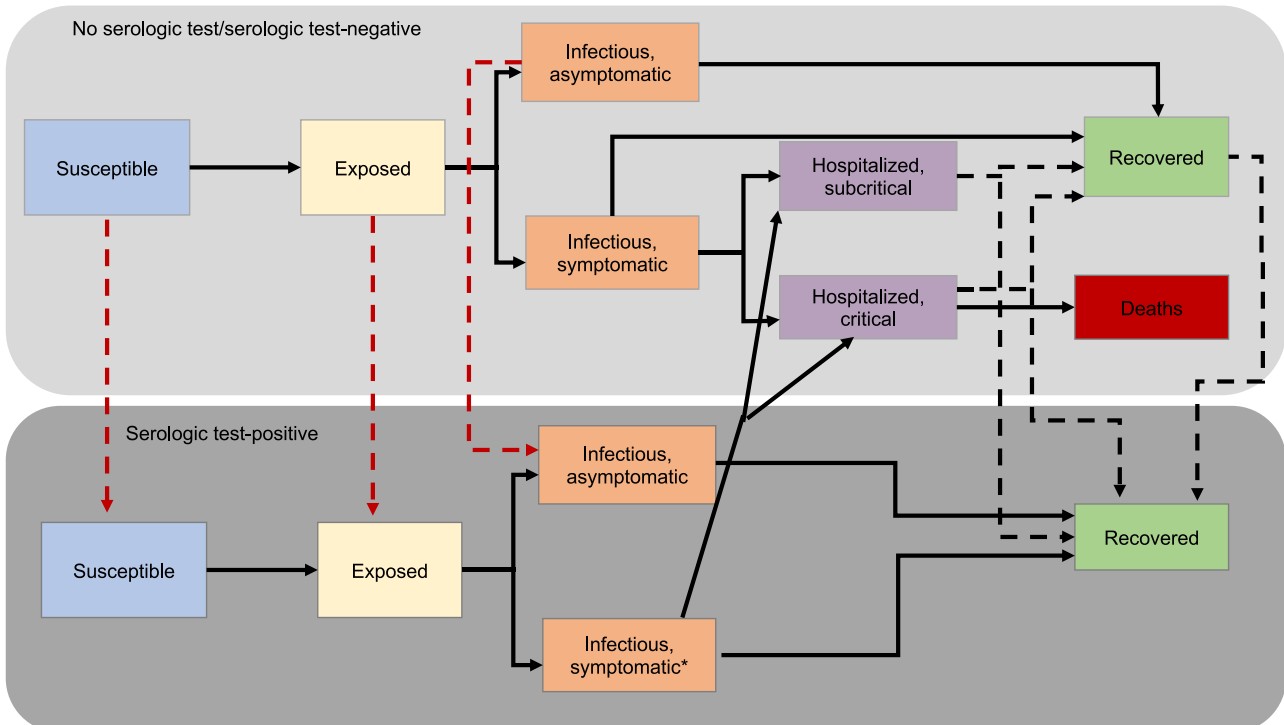

**Fig. 1 Overall model diagram.** Serological antibody testing is shown by dashed arrows. Red dashed arrows indicate either false positives (i.e., someone is not immune, but is moved to the test-positive group) and occur at a rate that is a function of 1-specificity, or false negatives (i.e., someone is recovered, but stays in the test-negative group). True positives occur at a rate that is a function of the sensitivity. The hospitalization compartments are located in the "Not tested/test-negative" layer for simplicity, though individuals who incorrectly test positive could move to these compartments after developing a symptomatic infection.

the consistency of fits across multiply-independently sampled chains implies that the model outcomes early in the epidemic are insensitive to variation in these parameters—enabling us to evaluate baseline predictions with and without serological testing.

**Epidemic dynamics in the absence of serological testing.** In all three sites, our models predicted a second epidemic peak in the fall and winter of 2020–2021, consistent with the qualitative shape of the epidemic trajectory (Fig. 3). For New York, the second peak is predicted to be smaller than the first, whereas the second wave is expected to be larger than the first wave in Washington and South Florida. If social distancing was sustained at Fall 2020 levels without any further interventions, our model predicts that 46–55% of the population across the three metropolitan areas (55% in New York City, 95% credible interval (CrI): 27–69%; 46% in South Florida, 95% CrI: 31–60%; and 46% in Washington, 95% CrI: 2–55%) would be infected with SARS-CoV-2 by June 2021, resulting in 72,000 cumulative deaths across the three sites (43,000 deaths in New York City, 95% CrI: 21,000–64,000; 10,000 deaths in South Florida, 95% CrI: 6000–17,000; and 19,000 deaths in Washington, 95% CrI: 1000–32,000) since the start of the pandemic (Fig. 4, top row). In reality, the death count for all three locations was 50,272 (34,492 in New York City, 12,729 in South Florida, 3051 in Washington)—within the 95% CrI in all cases.

**Epidemic dynamics with serological shielding.** Next, we retrospectively assess the benefit of a serological shielding strategy implemented in Fall 2020 in each metropolitan area, assuming that test-positive individuals increase their relative rate of interactions, thereby shielding susceptible individuals and reducing the risk of transmission. Specifically, individuals who test positive return to work and increase other contacts to normal levels. We

assume that test-negative and untested individuals continue to work from home if their job allows them to do so. To reflect the placement of test-positive individuals in high-contact roles, we assume that contacts at work and other (non-home, non-school) locations are preferentially with test-positive persons. When shielding interactions are 5:1 relative to that of those under social distancing guidelines, the probability of interacting with a test-positive individual is five times what would be expected given the frequency of test-positive individuals in the population, following the model of fixed shielding described in[10]. In each site, monthly serological testing of the population leads to a flattened epidemic curve in the fall and winter of 2020–2021. Widespread serological testing combined with moderate serologically-informed shielding (5:1) starting on November 1, 2020, using a highly specific test, could have reduced cumulative deaths by June 2021 by 22% across the three sites combined. The strongest reductions are in Washington (59%, 95% CrI for deaths averted: 0–17,000), with a lower relative impact in New York City (8%, 95% CrI for deaths averted: 300–600) and South Florida (14%, 95% CrI for deaths averted: 900–1300) (Fig. 4, top row).

**Impacts of serological testing frequency on epidemic outcomes and release from social distancing.** Simulations of test-based interventions reveal that the magnitude of the benefit from serological shielding depends on the frequency of testing, with more frequent testing resulting in both larger reductions in deaths and in a greater proportion of the population being released from social distancing if a highly specific test is used (Fig. 4, bottom row). In New York, monthly population testing would have been needed to maximize the potential benefit, leading to 51% of the population being released from social distancing by June 1, 2021 (95% CrI: 27–70%) and deaths being reduced by 3000 (95% CrI for total deaths: 20,000–63,000). In contrast, annual population

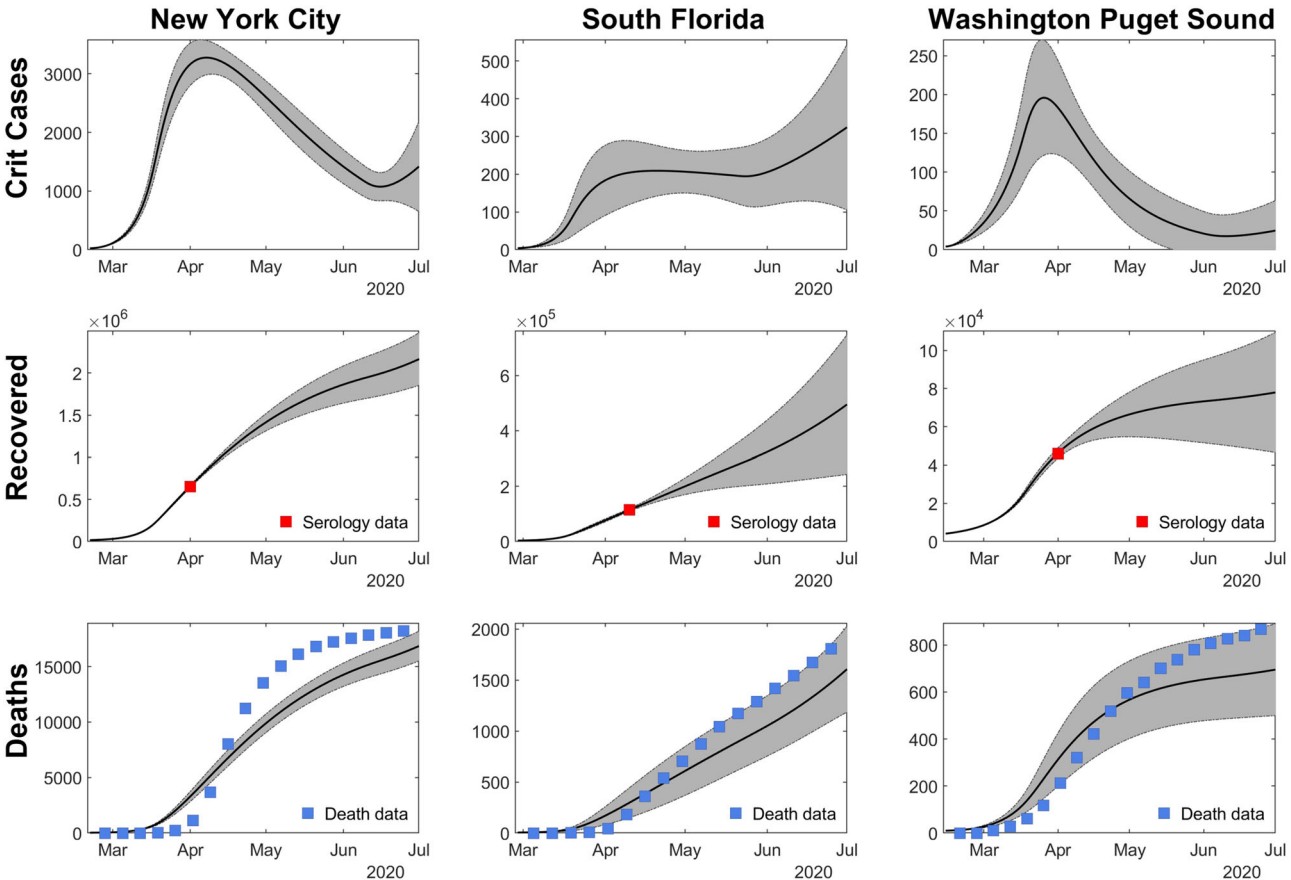

**Fig. 2 The first row shows the consistency between the fitted model and the deaths/seroprevalence data for New York City, South Florida, and Washington Puget Sound.** Daily critical care cases through July 1. The second row shows the cumulative number of recovered (previously infected) individuals. Red squares show the seroprevalence estimates from Havers et al. in each location[16]. In the third row, the cumulative deaths are shown, with death data shown in blue squares[24]. Data are presented as mean (black line) ±1.96 sd (gray bands), calculated from 100 random samples. Gray bands show 95% credible intervals, derived from the last 5000 iterations of converged MCMC chains.

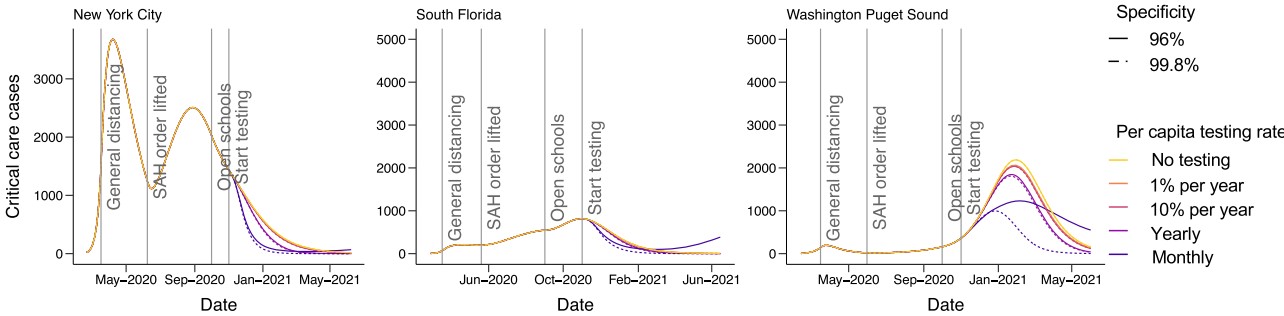

**Fig. 3 Critical care cases over time by testing level and location assuming 5:1 shielding.** Dates corresponding to the start of general social distancing in March 2020 and lifting at stay-at-home (SAH) orders in May and June 2020, are based on the dates that policies were enacted, or restrictions lifted, in each location. We assume that schools reopened at 50% capacity on September 1, 2020 in South Florida and October 1, 2020 in Washington and New York. Dotted lines show the impacts of a test with 90% specificity and solid lines show a test with 99.8% specificity. The 99.8% specificity scenario represents the accuracy reported among antibody tests currently authorized for use in the U.S., whereas the 90% specificity scenario is meant to capture reductions in accuracy that might be expected in a mass testing program.

testing would have been expected to release 26% of the population from social distancing (95% CrI: 13–37%) with 1500 deaths averted (95% CrI for total deaths: 20,000–64,000). More frequent testing would have also been beneficial in Washington; monthly testing would have released 21% of the population from social distancing (95% CrI: 3–33%) with 11,000 deaths averted (95% CrI for total deaths: 1000–15,000), compared with only 14% released

(95% CrI: 1–22%) and 4000 deaths averted (95% CrI for total deaths: 1000–26,000) with annual testing. In South Florida, 41% (95% CrI: 24–61%) of the population would have been released from social distancing with monthly testing, compared with 21% (95% CrI: 12–32%) with yearly testing. Monthly testing would have averted 1500 deaths in South Florida (95% CrI for deaths: 5000–15,000), whereas annual testing would have averted 500

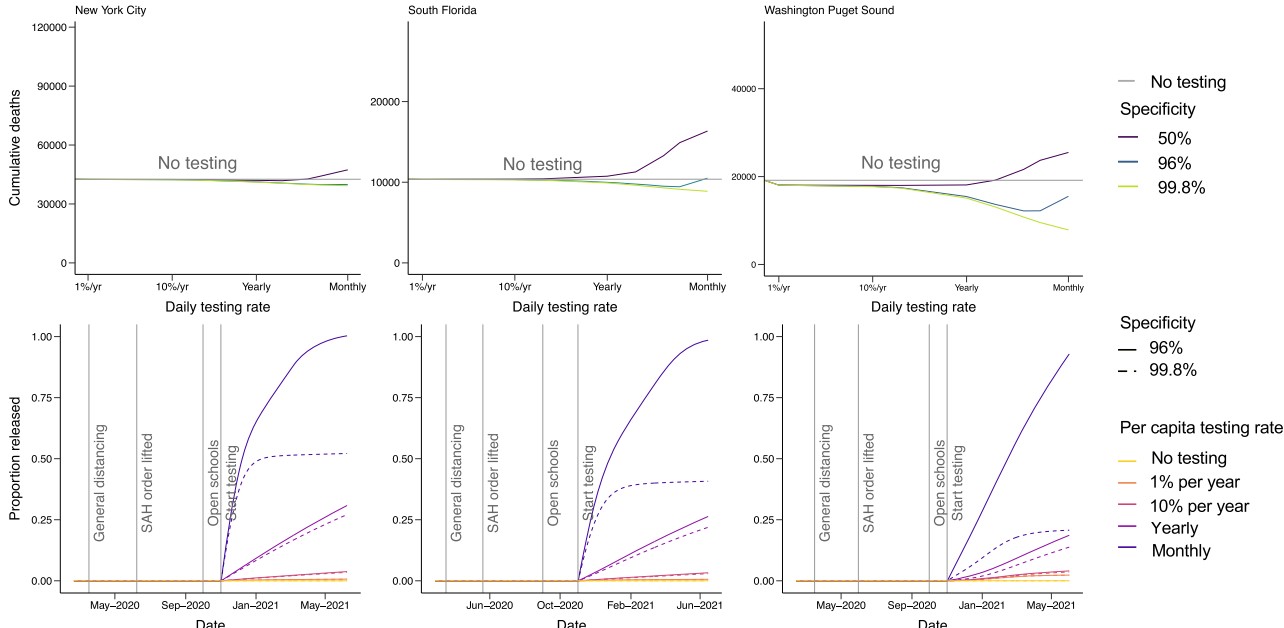

**Fig. 4 Cumulative deaths and number released from social distancing.** The top row shows cumulative deaths by location (panels) by daily testing rate from March 2020 to February 2021 for the scenario with 5:1 shielding, with schools reopening on September 1, 2020 in South Florida and October 1, 2020 in Washington and New York. Colored lines show test specificity. The gray horizontal line shows the number of deaths in the no-testing scenario for each location. The bottom row shows the fraction of the population of each metropolitan area released from social distancing by June 1, 2021, assuming 5:1 shielding. Line colors correspond to testing levels; blue is monthly testing (10 million tests/day) of the U.S. population. Dashed lines show expected results with a highly specific test (specificity = 99.8%) and solid lines show expected results with a test with 90% specificity. The 99.8% specificity scenario represents the accuracy reported among antibody tests currently authorized for use in the U.S., whereas the 90% specificity scenario is meant to capture reductions in accuracy that might result from the implementation of a mass testing program. The 50% specificity level represents a scenario in which an antibody test cannot distinguish between immune and non-immune individuals.

deaths (95% CrI for deaths: 6000–16,000). While increasing the intensity of social distancing toward the level of restrictions observed in April 2020 could help reduce deaths, these same benefits could be achieved by adding serological testing as part of a control strategy, allowing social distancing to be safely relaxed. As social distancing measures are relaxed further, testing frequency should also increase to minimize deaths and maximize the proportion of the population that can be released (Fig. 5). The extent to which testing frequency must increase to compensate for relaxing social distancing varies by location. For example, in New York City and South Florida, distancing could have been relaxed fully if monthly testing was employed.

**Impacts of serological testing performance and shielding on epidemic outcomes and release from social distancing.** The value and safety of a serological testing strategy depend on the level of shielding and test specificity. Thus far, our results centered on dynamics enabled by a high-performance test with a specificity of 99.8%, consistent with the high end of the range of reported specificity of available antibody tests[21]. We also explored the impact of employing a suboptimal test with 90% specificity, consistent with the lower range of approved tests plus additional decreases in accuracy due to rolling out testing at mass scale. Under this scenario, cumulative deaths across the three locations (66,000) would have been lower than if no-testing strategy was implemented (72,000) but higher than if using a high-performance test (56,000), with 93–99% of the population released from social distancing (New York: 99%, 95% CrI: 95–99%; South Florida: 98%, 95% CrI: 97–100%; Washington: 93%, 95% CrI: 73–97%). However, if monthly testing with a suboptimal assay (90% specificity) was implemented without

shielding, 97–99% of the population would have been released from social distancing (99% in New York City, 95% CrI: 95–99%; 98% in South Florida, 95% CrI: 97–100%; and 97% in Washington, 95% CrI: 73–99%) and 78,000 deaths would be expected, more than if no testing were implemented. Overall, adding shielding to a monthly testing strategy results in 10–27% fewer deaths compared to testing at the same frequency without shielding (10% in New York; 14% in South Florida; 27% in Washington). We also set test specificity to 50% to represent a scenario in which antibodies are not a reliable correlate of immunity (i.e., the test is poor at distinguishing between immune and non-immune individuals). If antibodies are not a reliable correlate of protection (which would be counter to current evidence that shows neutralizing antibodies persist for months[25]) then serological testing could lead to more deaths than if not used at all (Fig. 4, top panel). We conclude that shielding strategies avert deaths with any level of social distancing even when using a moderately specific test (90%) so long as antibodies provide a reasonably good correlate of protection (Fig. 5).

As a sensitivity analysis, we also explored how uncertainty in the natural history parameters (latent period, relative transmissibility of asymptomatic infections, hospital length of stay, and duration of symptomatic and asymptomatic infection) altered the impact of testing and shielding. In general, if asymptomatic cases are more able to transmit than we have assumed in our main model, the impact of shielding would be enhanced. In contrast, faster recovery rates for both symptomatic and asymptomatic cases could decrease the ultimate impact of shielding, particularly in South Florida and Washington, where the initial epidemic wave was relatively mild. Changing the latent period and the duration of hospitalization had minimal impact on the results (Supplementary Figs. 12–14).

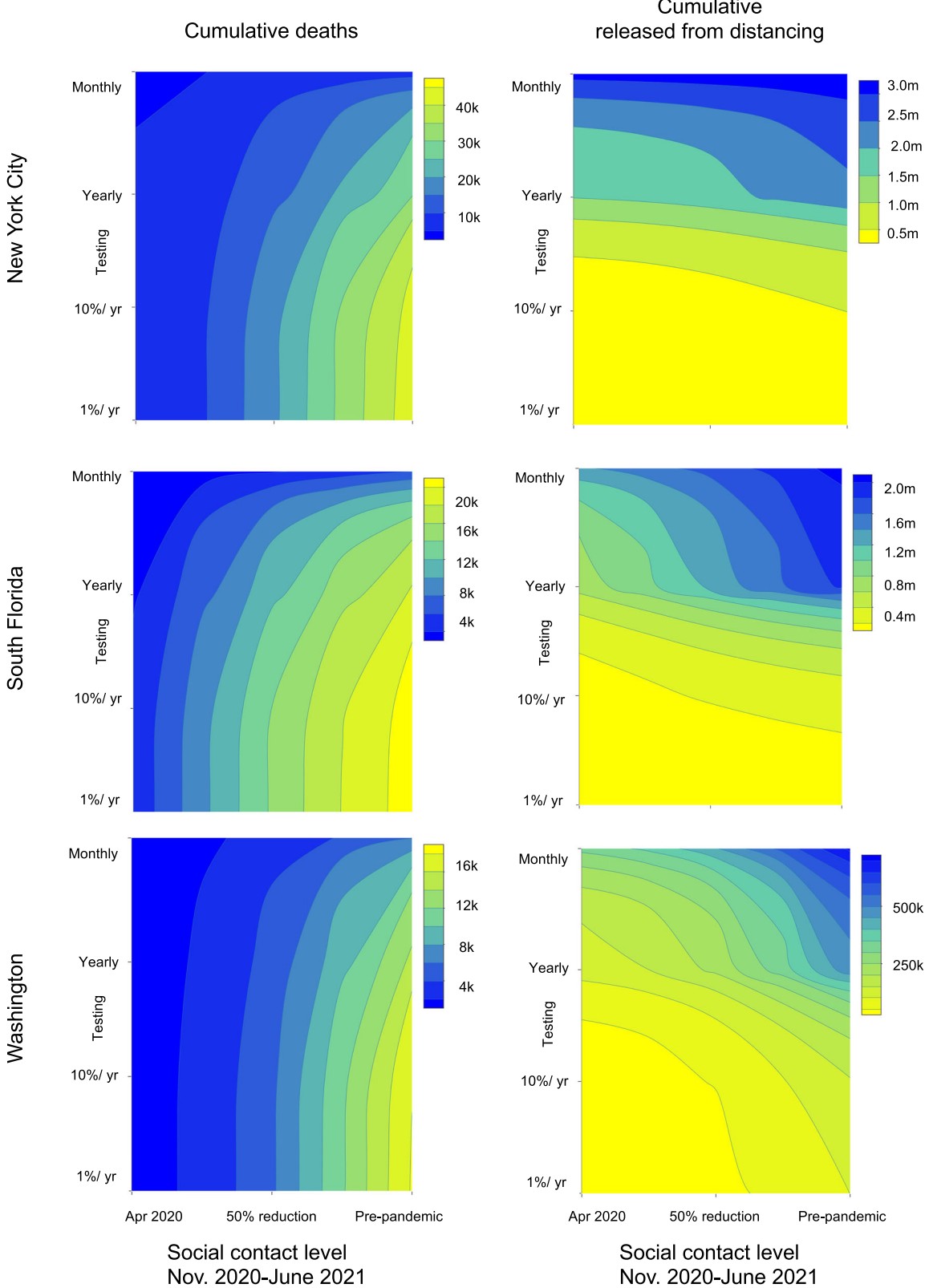

**Fig. 5 Cumulative deaths and number released from distancing by testing level and contact reductions.** Contour plot of cumulative deaths in each location from November 1, 2020 to June 1, 2021 (left column) and the number of people released from social distancing (right column) as a function of the degree of relaxation of social distancing and number of tests per day. The far right of the x-axis corresponds to a pre-pandemic level of contact and the far left corresponds to the contact levels in each location during stay-at-home orders in March–June 2020. Both panels assume a test specificity of 99.8% and a shielding factor of 5:1.

## Discussion

After achieving reasonably good fits of the model to historical data, our simulation study reveals that sufficiently frequent testing using high-performance tests combined with serological shielding in the pre-vaccine era would have decreased deaths and allowed relaxed social distancing for a substantial fraction of the population. First, we found that maintaining moderate social distancing equivalent to levels in Fall 2021, together with monthly serological testing, could have relieved 21–51% of several U.S. metropolitan populations from social distancing by June 2021. Second, if moderate shielding was employed, a strategy with serological testing would have resulted in up to 16,000 fewer deaths than a strategy without testing in the three focal areas. Adding shielding alongside monthly testing could have further reduced mortality rates and allowed a substantial fraction of the population to return to work and other activities with relative safety without the social and economic costs of strict, prolonged social distancing measures[26]. Third, we find that such a strategy could in fact prove dangerous, resulting in more deaths, if the serology test is non-specific or if antibodies are not a reliable indicator of immunity. Vaccine passports are already in place in some countries to identify individuals with immunity due to vaccination or recent history of infection[27]. While our models were fit to data before COVID-19 vaccines were widely available, the principle of serological shielding may still prove useful. For example, serological testing could be used to complement vaccination status, identifying more individuals with immunity for whom social distancing could be more safely relaxed.

An aggressive, monthly testing approach is unprecedented but, we argue, may be feasible and warranted under certain scenarios considering the continued social and economic impact of the epidemic. Implementation would require a significant and rapid scale-up of serological testing capacity. In the U.S., this scale-up was achieved for diagnostic PCR testing; the U.S. expanded testing from fewer than 1000 tests per day in early March to nearly 250,000 tests per day in mid-May and about 1 million per day by September. Moreover, recently developed serological tests are quicker to perform than RT-PCR[28,29]. New York City reported performing a peak of 187,000 tests per week in late March 2020[30], which corresponds to a rate between the yearly testing and 10% per year testing scenarios we consider in our analysis and could likely be increased with a concerted focus on antibody testing. Highly specific, self-administered bloodspot assays, as well as saliva-based tests[31,32], could ease some of the logistical challenges of large-scale testing.

Still, there is a legitimate concern that serological tests to relax social distancing could increase population risk[33]. To the contrary, we show that coupling serological testing using available diagnostic tests with immune shielding can form the basis of a successful risk mitigation strategy. If testing employs the most specific assays available, the false positive proportion would remain low and decrease over time as seroprevalence increases. If test specificity is closer to 90%, the false positive rate could have reached 50% in all sites while remaining lower than the prevalence of positives in the untested population. As such, deploying immune individuals such that they are responsible for more interactions than susceptible individuals will reduce risk. If shielding is not employed, this benefit disappears, and testing can become a liability—reinforcing the critical need to combine serological testing with a shielding strategy. Importantly, false positives are unlikely to substantively impact population-level risk at levels of specificity reported by most authorized serological tests[21,34].

In this modeling study, there are a number of assumptions and limitations that should be considered. First, while the credible intervals of cumulative mortality from our models overlap with realized outcomes, there remains considerable uncertainty regarding the extent to which individuals continued to practice social distancing through mid-2021, which is a key parameter in our models. This underscores the challenges of predicting the trajectory of the epidemic amidst uncertainty about shifting behavioral patterns. Social distancing was broadly adopted in the initial response in the United States[35] but quantifying ever-evolving patterns of social mixing is challenging and little empirical data on behavioral patterns starting in March 2020 is available. Nevertheless, modeling the impact of changing contact patterns on disease transmission is a critical aspect of our model.

Second, our models assumed random allocation of serological testing. In practice, targeting testing to specific groups, such as healthcare workers, nursing home care providers, food service employees, or contacts of confirmed or suspected cases might increase efficiency by increasing the test-positive rate (and consequently, cost-effectiveness[36]), allowing for similar numbers of individuals to be released from social distancing at lower testing levels. This strategy would also decrease the false positive rate, an important consideration if a less specific test is used[37]. Many healthcare organizations have already begun to offer antibody testing to their employees[38]. The use of serological testing and shielding within healthcare settings represents a smaller-scale, more targeted application of a testing and shielding strategy[39].

Third, we have made three critical simplifying assumptions in our model. We assume that antibodies are immediately detectable after resolution of infection. In reality, this generally occurs between 11 and 14 days post infection[40]. A small fraction of recent infections would be undetected, but this would likely have a minor effect on our results. Next, we assume that immunity lasts for the duration of our simulations, or at least 15 months. Both the duration of antibody protection and the extent to which those antibodies protect against future infections remains unclear. However, the vast majority of individuals who are infected seroconvert[18], and ongoing studies of SARS-CoV-2 show that antibodies persist for at least months[40]. Even as antibodies wane, this does not necessarily imply the loss of immune protection[41]. Ongoing studies are needed to determine whether these same patterns hold true for newly emerging variants. In addition, we assume that antibodies detected by serology are a correlate of protection. While antibody levels have been shown to wane after several months[42,43], especially for individuals with mild infection[25,41], protection following natural infection remains substantial[44], even when antibodies may be undetectable. Finally, we assume that serology data from Havers et al.[16] are representative of the metropolitan areas in which the studies were conducted. In reality, convenience sampling was used in each location, taking advantage of medical visits for other reasons. While these numbers might be biased, the direction of this potential bias is unclear and Havers et al. remains the best serological data available at the time of writing. We also assume that the age-specific case fatality rates are constant over time[45]. If the actual fatality rate declines over time, this may have led us to overestimate the number of deaths.

Even if testing can be scaled up, legal and ethical concerns remain. Requiring evidence of a positive test to return to activities may create strong incentives for individuals to misrepresent their immune status or intentionally infect themselves. However, this is less of a concern amid the widespread availability of vaccines. Nonetheless, a mass testing program must consider how such policies might enforce existing social disparities and guard against inequities in test availability[46,47]. Moreover, attention must be paid to the potential risk posed by re-infection, which is especially of concern with new variants.

We have focused our analysis on serological testing, using the principles of serological shielding to reduce risk of infection for

susceptible individuals, but this principle also applies to vaccination[48]. As of July 2021, three vaccines against SARS-CoV-2 are widely available throughout the United States[49] and over 68% of U.S. adults have received at least one dose of vaccine[7]. However, vaccination uptake varies geographically. In the United States, if a rebound in transmission occurs among unvaccinated individuals as variants of concern begin to become more widespread, vaccinated individuals and/or seropositive individuals could also be preferentially placed in high-contact positions to serve as immune "shields". This could allow transmission to be more controlled, even as social distancing interventions continue to be relaxed. If a novel strain emerges that escapes vaccine-derived and natural immunity, additional testing could identify individuals who have immunity against the escape variant for shielding to be a viable strategy. This strategy might be particularly beneficial in high-risk settings, such as healthcare or long-term care facilities.

A serological testing strategy could be one component of the continued public health response to COVID-19, alongside vaccination, viral testing, masking and contact tracing. Our results show that serological testing coupled with shielding could have mitigated the impacts of the COVID-19 pandemic while also allowing a substantial number of individuals to safely return to social interactions and economic activity, suggesting a future role for serological testing in the ongoing public health response to COVID-19 amid low vaccination coverage and the continuing threat of emergent SARS-CoV-2 variants.

## Methods

We modeled the transmission dynamics of SARS-CoV-2 using a deterministic, compartmental SEIR-like model (Fig. 1). We assume that after a latent period, infected individuals progress to either asymptomatic or symptomatic infection. A fraction of symptomatic cases are hospitalized, with a subset of those requiring critical care. Surviving cases, both asymptomatic and symptomatic, recover and are assumed to be immune to re-infection. All individuals who have not tested positive and are not currently experiencing symptoms of respiratory illness are eligible to be tested and all hospitalized cases are tested prior to discharge. Recovered individuals are moved to the test-positive group at a rate that is a function of test sensitivity. Susceptible, latently infected, and asymptomatic cases may falsely test positive and are moved to the test-positive group at a rate that is a function of test specificity. False positives may become infected, but the inaccuracy of their test result is not recognized unless they develop symptoms that are sufficiently severe to warrant hospitalization and health providers correctly diagnose COVID-19, overriding the history of a positive antibody test. The ordinary differential equations corresponding to this model are included in section SI Appendix, Section S1. All models were run in R (version 3.6.2) using the package deSolve. Translations of the baseline model are available in Matlab and Python. Fitting, estimation, and visualization of fits were implemented in MatLab R2019a and Python version 3.7.3. Code is available at https://github.com/lopmanlab/Serological_Shielding.

There are three age groups represented in the model: children and young adults (<20 years), working adults (20–64 years), and elderly (65+ years). We modeled age-specific mixing based on POLYMOD data adapted to the population structure in the United States[50,51]. Contacts in this survey were reported based on whether they occurred at home, school, work, or another location. All baseline social contact matrices were based on Prem et al.[51] and calculated to be symmetric using the symmetric = TRUE option when calling the contact_matrix function in the socialmixr R package (version 0.1.6).

General social distancing began on the day that stay-at-home orders were enacted in each location. Although adherence to these measures varied and is generally difficult to measure, we made several assumptions about how these policies changed location-specific contacts. First, we assume that under these measures, all contacts at school were eliminated and that contacts outside of home, work, and school (other) locations were reduced by a fraction, which was fitted for each location. We assume that contacts at home remained unchanged. To address differences in work-based contacts by occupation types, we classified the working adult population into three subgroups based on occupation: (i) those with occupations that enable them to work exclusively from home during social distancing, (ii) those continuing to work but reduced their contacts at work (e.g., customer-facing occupations such as retail), and (iii) those continuing to work with no change in their contact patterns (e.g., frontline healthcare workers). The percent reduction in other contacts and percent contact reductions at work for essential workers who could reduce their contacts was fitted (see next section).

This period of intense social distancing lasts until stay-at-home orders are lifted in each location. All three municipalities enacted social distancing regulations in

mid-March 2020[52–54]. Under these measures, we assume all contacts at school were eliminated and contacts outside of home, work, and school (other) locations were substantially reduced[55] while contacts at home remained unchanged, with distancing starting at the time that stay-at-home orders were enacted in each site. After reopening begins, we assume that schools remain closed but that social distancing measures for the general population can be relaxed, by allowing work and other contacts to be increased. In accordance with school reopening policies in each location, we assume that schools remained closed until September 1, 2020 in South Florida and October 1, 2020 in Washington and New York. To represent general relaxation of social distancing, we scale contacts at work and other locations to a proportion of their value under general social distancing based on a scalar constant, $c$, such that $c = 1$ is equivalent to the scenario in which social distancing measures as put into place in March are maintained and $c = 0$ is equivalent to a return to pre-pandemic contact levels for both work and other contacts for essential workers and pre-pandemic contact levels for other contacts for all other groups. Based on local policies, we assume that children returned to school on September 1, 2020 in South Florida and October 1, 2020 in Washington and New York City. To account for the fact that schools have taken a variety of measures to reduce contact among students, we assumed that children halved (50%) their pre-pandemic contacts at school.

**Nonlinear model-data fitting**. We fit the model for each location to deaths reported due to COVID-19 from March to July 2020[24] as well as seroprevalence data[16] using a Markov Chain Monte Carlo (MCMC) approach[56,57] using the MCMCstat toolbox (https://mjlaine.github.io/mcmcstat/)[58,59]. Each location was defined using the same counties as were included in a CDC-led seroprevalence study[16]. For each location, we estimated the six parameters listed in SI Appendix. Model parameter values are shown in SI Appendix, Tables S1 and S2. Reductions in social contacts corresponding to these fitted parameter values are shown in SI Appendix, Table S3. We first performed Latin Hypercube Sampling to generate random parameter sets. Using each set, we ran initial fits and started our MCMC runs from the ten first sets associated with the minimum errors. We ran these ten randomly seeded chains for 100,000 iterations each (95,000 burn-in; 5,000 samples). To infer the initial conditions, we first calculated the number of weeks between the first reported death in each location and the first week where the cumulative death toll exceeded 10. Using region-specific conditions (population demographics, stay-at-home order enactment and lifting dates, and death data), we initialize an epidemic consisting of a single exposed adult, forward simulated until a death count threshold was met, and then the population distribution was used as the initial condition for subsequent intervention scenarios.

We used a Poisson likelihood function that included penalty terms for the cumulative midpoint and final number of deaths in each location, the weekly death rates, and population-level seroprevalence estimated for each location from Havers et al.[16]. We estimated chain convergence using the Gelman–Rubin diagnostic (Supplementary Fig 1). Supplementary Figs. 2, 5, and 8 show the trace plots for each model, and Supplementary Figs. 3, 6, and 9 show the resulting joint distributions of estimated parameters. The consistency between the fitted model and death/seroprevalence data for each location is shown in Supplementary Figs. 4, 7, and 10. After fitting to death data spanning March to July 2020, we use fitted parameters to forward-simulate the epidemic through June 1, 2021 in each location.

Given that all ten site-specific chains converged to similar values for each location (indicating good and consistent model fits), we randomly sampled 20 parameter sets in the final 5000 interations in each site for each location to capture uncertainty in model predictions. This resulted in 200 randomly sampled parameter sets for each site. We simulated the epidemic forward using each parameter set for the key testing and shielding interventions we report in the text. We report the middle 95% of the distribution of outcomes from these runs as our credible intervals. As only the fitted parameters were varied, the resulting uncertainty intervals only capture uncertainty in the fitted parameters and not in parameters that were fixed from prior literature. If all parameters had been varied, our credible intervals would likely have been wider. We have uploaded a supplementary file with the number of deaths, critical care cases, cumulative incidence, and the fraction of the population released from social distancing after one year from each of these simulations as Supplementary Material. More details regarding model fitting are shown in SI Appendix Sect. S3.

**Sensitivity analysis of fixed parameters using partial-rank correlation coefficients**. To capture the potential influence of uncertainty in fixed parameters on the impact of shielding, we used Latin Hypercube Sampling to generate 300 random parameter sets based on probable ranges for each parameter, assuming a uniform distribution within each range[60]. We varied the duration of the latent period from 3 to 12 days[61,62], the relative transmissibility of asymptomatic infection (compared to symptomatic infection) from 25 to 100%[63–66], the recovery for non-hospitalized, symptomatic cases from 1 to 10 days[67,68] and for asymptomatic cases from 3 to 8 days[69], and the length of hospitalization from for severe cases from 6 to 20 days and for non-severe cases from 3 to 9 days[70,71]. We sampled parameters separately from each distribution and simulated our main shielding scenarios for each. For each parameter, we calculated the partial-rank correlation coefficient between the value of the parameter and the number of deaths in the

simulation run with that parameter value. This provides a measure of the effect of the value of each fixed parameter on the impact of shielding in our models.

**Reporting summary**. Further information on research design is available in the Nature Research Reporting Summary linked to this article.

## Data availability

Death data used for model calibration are available from https://usafacts.org/visualizations/coronavirus-covid-19-spread-map/ and policy data that was used to define the reopening policies by the site are available at https://github.com/nytimes/covid-19-data. Seroprevalence data was previously published by Havers et al.[16]. All data used directly in model fitting are available in the Github repository, specifically at https://github.com/lopmanlab/Serological_Shielding/MCMC_CODE/Matlab/INPUTS.

## Code availability

Code is available at https://github.com/lopmanlab/Serological_Shielding[72].

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

## Acknowledgements

We thank Timothy Lash, Andreas Handel, Carly Adams, Julia Baker, Carol Liu, and Avnika Amin for useful comments on earlier versions of the manuscript. B.A.L. and A.N.M.K. were supported by the Vaccine Impact Modelling Consortium; B.A.L. and K.N.N. were supported by NIH/NICHD R01 HD097175; B.A.L., K.N.N., and A.N.M.K. were supported by NIH/NIGMS R01 GM124280; J.S.W. and D.D. were supported by Simons Foundation (Scope Award ID 329108); B.A.L. was supported by NSF 2032084 and NIH/NIGMS R01GM124280/GM124280-03S1; J.S.W. was supported by the Army Research Office (W911NF-19-1-0384); J.S.W. and C.Y.Z. were supported by National Science Foundation (2032082) and J.S.W. was supported by National Science Foundation (1806606, 1829636).

## Author contributions

A.N.M., K.N.N., J.S.W., and B.A.L. designed the study. The model was designed by A.N.M. and K.N.N., extended from an earlier version by J.S.W., D.D., and C.Y.Z. All authors designed model simulations, A.N.M., K.N.N., and C.Y.Z. conducted the analysis with input from D.D., B.A.L., and J.S.W.; D.D and C.Y.Z. led the model-fitting and A.N.M., K.N.N., and B.A.L. wrote the first draft of the manuscript. All authors contributed to editing the manuscript. J.S.W., D.D., and C.Y.Z. provided critical review of the code, results, and conclusions.

## Competing interests

B.A.L. reports grants and personal fees from Takeda Pharmaceuticals and personal fees from World Health Organization outside the submitted work. A.N.M., K.N.N., J.S.W., C.Y.Z., and D.D. declare no competing interests.
