## [Peer Review File · Nature Communications]

Reviewers' Comments:

Reviewer #2:

Remarks to the Author:

In this manuscript, Alicia et al established a transmission model, presented the data that proved serology can be implemented to allow seropositive individuals to increase levels of social interaction while offsetting transmission risks. There were kinds of SARS-CoV-2 antibody testing, the variation in different method may also limit the conclusion presented here. In addition, vaccination was also executed in population level, this strategy should more reliable to solve issues. In all, there were too many publications discussed the function of serological testing in 'immune shield', the study lacked of the novelty and found little new insights at present time.

Reviewer #3:

Remarks to the Author:

This paper applies a deterministic epidemiological model to explore the potential use of routine serological testing to implement a shield immunity strategy and thus allow for the relaxing of social distancing measures to reduce the transmission of Covid-19. The paper extends a published conceptual model by incorporating more detailed demographic structure stratifying contacts by age and location and calibrates the model to case notification and serological survey data from three geographic locations in the USA.

They key contributions of the study are demonstrating the critical role played by the diagnostic specificity of serological tests and the use of a calibrated transmission model to contextualise the quantitative trade-offs between the potential benefits in terms of the number of individuals released from restrictions against the costs in terms of expected mortality.

In order to achieve this the model makes several simplifying assumptions - in particular with respect to assuming that mixing is well mixed within risk groups and testing is both random (with respect to selection of individuals) and continuous (modelled as a continuous rate). While an individual based network model would be more appropriate to address the practical use of such a strategy - given the available data these approximation are pragmatic and the approach sensible. These limitations are carefully described along with the ethical and practical limitations of any shield immunity strategy.

Given that the model calibration is one of the key strengths of this paper, there are a number of concerning omissions in the reporting of the methodology which would make reproducibility difficult. Although source code is linked, key information on the estimation method is not reported in the manuscript. Furthermore the model has a very large number of fixed compared to estimated parameters. Together with the deterministic structure of the model this means that the model hugely underestimates the uncertainty in model predictions. This limitation is noted in the manuscript, but could have been mitigated substantially through using (less) informative prior distributions for key epidemiological quantities (such as the relative infectiousness of asymptomatic infections) which are currently fixed and likely to have a large impact on the impact of interventions.

Perhaps the biggest benefit - and primary motivation - for the use of Bayesian methods is to quantify the uncertainty of model estimates and predictions and make sensitivity analyses more efficient by reducing the range of parameter space necessary to explore. In this case prediction intervals are so small that sampling from the Bayesian posterior distributions provides hardly any additional information beyond simply simulating from the maximum likelihood values. As such, although it goes somewhat against the spirit of a fully Bayesian approach, there is a real need for a systematic sensitivity analysis to at least a subset of the fixed parameters to explore what impact they have on model predictions and the authors conclusions.

Specific Comments

1) Line 43: to avoid doubt would be helpful to qualify date is November 9 2020.

2) Line 142: The narrow credible intervals could (and I would argue) also be a symptom of the lack of realistic variability in the model (both the deterministic approximation and highly restrictive prior assumption on fixed transmission parameters).

3) Line 230: Would be useful to qualify this is not just a "moderately specific test" but at the low end of specificity estimates for real-world tests (would be helpful to reiterate this rationale in figure legends as well). Picking just three values to compare otherwise seems a little arbitrary and raises the additional question as to what the break-even specificity is. How good does a test have to be to eliminate the risk of negative impacts?

4) Line 266: Red line does not appear to be visible on plot?

5) Line 411: The source code repository has code written in Matlab and Python as well as R. In particular the MCMC code used appears to be in Matlab (with a corresponding Matlab implementation of the model). This should be clarified.

6) Line 419: There is far from a consensus on methods used to make contact matrices symmetric. For completeness would help to have specific method used in socialmixr R package presented here (as it may change in later revisions of the package).

7) Line 452: Again MCMCstat is Matlab software, is Matlab required to replicate this work?

8) Line 465: Would prior distributions not be a more natural way to do this than a penalised likelihood? Form and justification for likelihood should be given (at the very least in supplementary material). Likewise no prior distributions for estimated parameters are given - I assume that they are uniform on a fixed range but this is not clearly stated anywhere in the manuscript (making interpretation of the posterior distributions in the supplementary information almost impossible)

9) Line 476: As the Gelman-Rubin diagnostic is based on comparing multiple chains, how can only one chain have failed to converge? Surely the conclusion is that convergence test for all chains has failed? Do you routinely get convergence problems for sets of 10 chains? As presented this raises the concern that there are convergence issues for this data set and at the very least suggests individual chains need to be run for longer.

10) Line 662: Is the secondary New York Times aggregation really appropriate to use as a data source? Are official public sources available that could be used/cited? How trustworthy is the New York Times aggregation compared to official public data sources?

11) Supplementary information S5 really needs to be integrated within S1 as almost all of the estimated transmission parameters are only described here and absent from the model equations which is confusing. Likewise the notation using whole (or contractions) of words as variable names and subsets is extremely confusing. I understand the desire to make these variable names clearer to a non-mathematician but given the (relative) complexity of the model it makes the equations much more cumbersome than they need to be).

RESPONSE TO REVIEWER COMMENTS

In the text below, Reviewer's comments are in plain text and the Author's responses are in *italics*.

Authors note: In responding to these reviewer comments, we noticed an error in our initial model fitting, where the death rates had been specified incorrectly (albeit only with small, quantitative differences). We corrected this error and re-fit the model and re-ran all forward model simulations. Accordingly, all tables and figures have been updated. This small change has had no substantive impact on either the previous or current conclusions.

Reviewer #2 (Remarks to the Author):

In this manuscript, Alicia et al established a transmission model, presented the data that proved serology can be implemented to allow seropositive individuals to increase levels of social interaction while offsetting transmission risks. There were kinds of SARS-CoV-2 antibody testing, the variation in different method may also limit the conclusion presented here. In addition, vaccination was also executed in population level, this strategy should more reliable to solve issues. In all, there were too many publications discussed the function of serological testing in "immune shield", the study lacked of the novelty and found little new insights at present time.

This manuscript presents the potential impacts of a novel intervention, "immune shielding" that remains relevant as we move into a new phase of the pandemic, where individuals are becoming protected either through prior infection or vaccination, and social distancing measures in place for the last year begin to be relaxed. The epidemiologic implications of this strategy for returning to normal levels of social and economic activity have not been examined. Our analysis provides insight into the context-specific potential benefits of such a strategy given the accuracy of current antibody tests for SARS-CoV-2.

In the new version of the manuscript, we have included results adding vaccination to our model in the Supplemental Materials (Supplemental Table S4). In this extension of our main model, we assume a constant complete vaccination rate of 0.5% of the population per day (similar to the US peak distribution of 3 million doses/day when accounting for lower rates due to the need to administer 2 doses to complete the series) starting on January 1, 2021. We assume a vaccine efficacy of 95%: vaccinated, protected individuals (95% of vaccinees) are moved from the untested compartments to the tested, recovered (R) compartment and vaccinated, unprotected persons (5% of vaccinees), are placed in the tested, susceptible (S) compartment. This represents an "all-or-nothing" vaccine, whereby vaccinated persons receive complete protection from SARS-CoV-2 infection and disease. Interestingly, we find that the impact on overall deaths is not large relative to shielding if vaccination is implemented at a rate similar to that seen currently (and surmise that less effective vaccines will also lead to modest reductions in overall fatalities). While vaccines are undoubtedly highly effective at mitigating pandemic effects, this result underscores the large effect of implementing a testing and shielding intervention on a population-level. On top of the impact of a shielding strategy, vaccination provides little additional benefit. Of note, in New York City, where the benefit of shielding is very sensitive to small changes in test specificity (Figure 4), our scenarios with vaccination lead to slightly more deaths than those with shielding alone, since a vaccine with < 100% efficacy leads to 'false positives', and thus increased risk because those who are vaccinated but unprotected resume pre-pandemic levels of social interaction but can still be infected and transmit).

Location	Deaths with vaccination + shielding	Deaths with shielding only	Deaths with no intervention
New York City	3505	1766	5069
South Florida	2302	2327	3818
Washington	4256	6233	17528

Table S4. Deaths from November 1, 2020-June 1, 2021 under each scenario when adding vaccination to the model. We assume a shielding strategy of monthly testing with a highly-specific test ($sp = 0.998$) and 5:1 shielding that starts on November 1, 2020

Of course, vaccinations are now widely available, and both vaccination and prior natural infection contribute to current population-level immunity to SARS-CoV-2. In the discussion, we have now highlighted the potential applications of a shielding strategy at the current stage of the pandemic, which may include implementation in high-risk settings in the U.S, amidst waning immunity from both vaccines and prior infection, and in countries with limited access to SARS-CoV-2 vaccines.

We agree the variation in specificity of antibody tests will change the expected impacts of a widespread serological testing strategy. While our initial manuscript tested three different levels of specificity (99.8%, 98%, and 95%) based on the distribution of reported accuracy of tests that have received Emergency Use Authorization (EUA) by the Food and Drug Administration (FDA)(1), we now also present results for 90% specificity. While this is lower than any of the tests that have received EUAs, we recognize that the programmatic usage of a test may result in lower specificity than the ideal conditions under which sensitivity and specificity and were ostensibly tested prior to receiving EUA.

Reviewer #3 (Remarks to the Author):

This paper applies a deterministic epidemiological model to explore the potential use of routine serological testing to implement a shield immunity strategy and thus allow for the relaxing of social distancing measures to reduce the transmission of Covid-19. The paper extends a published conceptual model by incorporating more detailed demographic structure stratifying contacts by age and location and calibrates the model to case notification and serological survey data from three geographic locations in the USA.

The key contributions of the study are demonstrating the critical role played by the diagnostic specificity of serological tests and the use of a calibrated transmission model to contextualise the quantitative trade-offs between the potential benefits in terms of the number of individuals released from restrictions against the costs in terms of expected mortality.

In order to achieve this the model makes several simplifying assumptions - in particular with respect to assuming that mixing is well mixed within risk groups and testing is both random (with respect to selection of individuals) and continuous (modelled as a continuous rate). While an individual based network model would be more appropriate to address the practical use of such a strategy - given the available data these approximations are pragmatic and the approach sensible. These limitations are carefully described along with the ethical and practical limitations of any shield immunity strategy.

Given that the model calibration is one of the key strengths of this paper, there are a number of concerning omissions in the reporting of the methodology which would make reproducibility difficult. Although source code is linked, key information on the estimation method is not reported in the manuscript. Furthermore, the model has a very large number of fixed compared to estimated parameters. Together with the deterministic structure of the model this means that the model hugely underestimates the uncertainty in model predictions. This limitation is noted in the manuscript, but could have been mitigated substantially through using (less) informative prior distributions for key epidemiological quantities (such as the relative infectiousness of asymptomatic infections) which are currently fixed and likely to have a large impact on the impact of interventions.

Perhaps the biggest benefit - and primary motivation - for the use of Bayesian methods is to quantify the uncertainty of model estimates and predictions and make sensitivity analyses more efficient by reducing the range of parameter space necessary to explore. In this case prediction intervals are so small that sampling from the Bayesian posterior distributions provides hardly any additional information beyond simply simulating from the maximum likelihood values. As such, although it goes somewhat against the spirit of a fully Bayesian approach, there is a real need for a systematic sensitivity analysis to at least a subset of the fixed parameters to explore what impact they have on model predictions and the authors conclusions.

Thank you for these thoughtful and constructive comments. In the new version of the manuscript, we have expanded our description of the estimation methods used (see responses to specific comments below).

We agree with the reviewer that our results may be sensitive to our assumptions about key parameters and that this uncertainty is not represented in uncertainty intervals presented. However, identifiability concerns prevented us from fitting additional parameters in the model. Nonetheless we agree that it is important to test the sensitivity of our main results to the values we assumed for key fixed parameters. In the revised version of the manuscript, we have included a sensitivity analysis using partial rank correlation coefficients of key fixed parameters (2,3), including the relative infectiousness of asymptomatic cases, the latent (incubation) period, recovery rates for both symptomatic and asymptomatic infections, and duration of hospitalization (Supplemental Figures S12-S14). In general, increased relative transmissibility for asymptomatic cases tended to increase the potential for deaths averted through testing/shielding, whereas faster recovery rates for both symptomatic and asymptomatic cases tended to decrease the potential impact of shielding. These patterns were most pronounced for South Florida and Washington. In New York City, a shorter latent period was associated with decreased potential impact of shielding when monthly testing with a highly specific test was used, but not in any of the other intervention scenarios considered. These patterns were reversed when testing was employed without shielding. The duration of hospitalization did not alter the ultimate impact of a shielding strategy in any of the sites.

Specific Comments

- 1) Line 43: to avoid doubt would be helpful to qualify date is November 9, 2020.

We have added years to all dates in the text.

- 2) Line 142: The narrow credible intervals could (and I would argue) also be a symptom of the lack of realistic variability in the model (both the deterministic approximation and highly restrictive prior assumption on fixed transmission parameters).

See our response to the main critique above. We have included a new sensitivity analysis that addresses this point.

- 3) Line 230: Would be useful to qualify this is not just a "moderately specific test" but at the low end of specificity estimates for real-world tests (would be helpful to reiterate this rationale in figure legends as well). Picking just three values to compare otherwise seems a little arbitrary and raises the additional question as to what the break-even specificity is. How good does a test have to be to eliminate the risk of negative impacts?

The three values were based on the distribution of specificity reported by tests that have received Emergency Use Authorization (EUA) for use by the U.S. FDA. We have reiterated this rationale in the Figure legends but in the new version of the manuscript have also included a substantially less specific test (90% specificity), in order to account for the fact that actual specificity is likely to be lower in the context of a widespread testing initiative than under initial evaluation conditions. Using a test with 90% specificity, we found that increases in serological testing can provide benefits in terms of deaths averted when used with any level of social distancing so long as shielding is also used. The 'break-even specificity' will depend on the setting-specific epidemic state: for example, based on Figure 2, a 90% specific test could be used in New York City and South Florida without causing additional deaths, but a lower-specificity test could be used in Washington.

- 4) Line 266: Red line does not appear to be visible on plot?

Fixed.

- 5) Line 411: The source code repository has code written in Matlab and Python as well as R. In particular the MCMC code used appears to be in Matlab (with a corresponding Matlab implementation of the model). This should be clarified.

We have clarified the languages in which different parts of the model and estimation procedures are implemented in the Materials and Methods. Revised text is shown below:

All models were run in R version 3.6.2 using the package 'deSolve'. Translations of the baseline model are available in Matlab and Python. Fitting and estimation were implemented in MatLab R2019a. Code is available at https://github.com/lopmanlab/Serological_Shielding.

- 6) Line 419: There is far from a consensus on methods used to make contact matrices symmetric. For completeness would help to have specific method used in socialmixr R package presented here (as it may change in later revisions of the package).

We have clarified the version of socialmixr package that we used and the function that we used to symmetrize matrices.

- 7) Line 452: Again MCMCstat is Matlab software, is Matlab required to replicate this work?

Yes, this has been clarified in the Methods and the SI.

- 8) Line 465: Would prior distributions not be a more natural way to do this than a penalised likelihood? Form and justification for likelihood should be given (at the very least in supplementary material). Likewise no prior distributions for estimated parameters are given - I assume that they are uniform on a fixed range but this is not clearly stated anywhere in the manuscript (making interpretation of the posterior distributions in the supplementary information almost impossible)

Yes, prior distributions were uniform on a fixed range. We have added the prior distributions used to fit parameters to the SI for completeness as well as the log-likelihood function.

We were unable to find reliable information to use for prior distributions for each of the locations in our model and opted for a more interpretable likelihood penalization approach. We did constrain the ranges of parameters used in the fitting based on current literature, but used a uniform prior distribution to appropriately reflect our uncertainty. The plausible ranges for each parameter have been added to the supplemental material. This is a common approach for MCMC methods.

The likelihood function is the sum of the Poisson log-likelihoods. Let $d_o(t)$ be the observed cumulative number of deaths at time t and $r_o(t)$ be the observed weekly death rate. Let $d_s(t)$ be the model simulated cumulative number of deaths, and $r_s(t)$ be the model simulated weekly death rates at time t . Let σ_s and σ_o denote the simulated and observed population-level

seroprevalence estimated for each location by Havers et al, $a = 100000$ be a scaling factor, and F_{lp} denote the Poisson log-likelihood function.

Then, our log-likelihood function is defined as:

$$LL = \sum_t \left(F_{lp} \left(\frac{r_s(t)}{a}, \frac{r_o(t)}{a} \right) \right) + F_{lp} \left(\frac{d_s(t_m)}{a}, \frac{d_o(t_m)}{a} \right) + F_{lp} \left(\frac{d_s(t_f)}{a}, \frac{d_o(t_f)}{a} \right) + F_{lp} \left(\frac{\sigma_s}{a}, \frac{\sigma_o}{a} \right)$$

- 9) Line 476: As the Gelman-Ruben diagnostic is based on comparing multiple chains, how can only one chain have failed to converge? Surely the conclusion is that convergence test for all chains has failed? Do you routinely get convergence problems for sets of 10 chains? As presented this raises the concern that there are convergence issues for this data set and at the very least suggests individual chains need to be run for longer.

We have re-fit the model for this revision (see author's note), and using our current approach to estimate 5 parameters, all chains converged. Fitting two additional parameters (relative transmissibility of asymptomatic infection and the latent period) yielded poor convergence and seemed to cause additional identifiability concerns, so we could not estimate additional parameters for each site with confidence (see Supplemental Figure S1, which shows poor convergence of the 7-parameter fits). Each fit was run for 10 chains and 100,000 iterations (95,000 burn-in; 5,000 sample). Based on these results, we elected to use the 5-parameter fits for all three sites and to explore the potential influence of other fixed model parameters using Latin Hypercube Sampling (discussed in response above under the general comments).

- 10) Line 662: Is the secondary New York Times aggregation really appropriate to use as a data source? Are official public sources available that could be used/cited? How trustworthy is the New York Times aggregation compared to official public data sources?

The time series of deaths used to calibrate the transmission model was taken from USA Facts. We have updated our data availability statement as the COVID Tracking Data source was a typo in our original submission. The USA facts database is cited in reference 24 in the main text. While death data were available from other sources besides USA Facts (such as the COVID tracking project), many sources only provided these data at the state level early in the pandemic. While the COVID tracking project has these data from May-October 2020, data are unavailable from the first few months of the pandemic, which were used for model calibration. Given that the US epidemic has had dramatically different dynamics that vary even by city within a state, we did not want to use death data that were aggregated to a lower level of resolution. The only data from the New York Times that were used were for the reopening policies in each state, which was used as an initial condition for the forward simulations but did not influence the model fitting.

- 11) Supplementary information S5 really needs to be integrated within S1 as almost all of the estimated transmission parameters are only described here and absent from the model equations which is confusing. Likewise the notation using whole (or contractions) of words as variable names and subsets is extremely confusing. I understand the desire to is to make these

variable names clearer to a non-mathematician but given the (relative) complexity of the model it makes the equations much more cumbersome than they need to be).

We have included former section S5 immediately after S1 and have changed the variable names in the SI to make the equations more readable.

References:

1. Health C for D and R. EUA Authorized Serology Test Performance. FDA [Internet]. 2020 May 7 [cited 2020 May 11]; Available from: <https://www.fda.gov/medical-devices/emergency-situations-medical-devices/eua-authorized-serology-test-performance>
2. Marino S, Hogue IB, Ray CJ, Kirschner DE. A Methodology For Performing Global Uncertainty And Sensitivity Analysis In Systems Biology. *J Theor Biol.* 2008 Sep 7;254(1):178–96.
3. Wu J, Dhingra R, Gambhir M, Remais JV. Sensitivity analysis of infectious disease models: methods, advances and their application. *J R Soc Interface.* 2013 Sep 6;10(86):20121018.

Reviewers' Comments:

Reviewer #2:

Remarks to the Author:

The revised manuscript included results adding vaccination to model and the discussion about performance of antibody tests, it's great improved in my personal opinion.

In discussion section (Line 364), when authors mentioned 'antibody levels have been shown to wane after several months', 2 references should be cited for their description of antibody wane in early time.

1) Long QX, Tang XJ, Shi QL, Li Q, Deng HJ, Yuan J, Hu JL, Xu W, Zhang Y, Lv FJ, Su K, Zhang F, Gong J, Wu B, Liu XM, Li JJ, Qiu JF, Chen J, Huang AL. Clinical and immunological assessment of asymptomatic SARS-CoV-2 infections. *Nat Med*. 2020 Aug;26(8):1200-1204. doi: 10.1038/s41591-020-0965-6. Epub 2020 Jun 18. PMID: 32555424.

2) Ibarrodo FJ, Fulcher JA, Goodman-Meza D, Elliott J, Hofmann C, Hausner MA, Ferbas KG, Tobin NH, Aldrovandi GM, Yang OO. Rapid Decay of Anti-SARS-CoV-2 Antibodies in Persons with Mild Covid-19. *N Engl J Med*. 2020 Sep 10;383(11):1085-1087. doi: 10.1056/NEJMc2025179. Epub 2020 Jul 21. Erratum in: *N Engl J Med*. 2020 Jul 23;: PMID: 32706954; PMCID: PMC7397184.

Reviewer #3:

Remarks to the Author:

First I would like to thank the authors for the comprehensive and detailed response to the questions and queries in my original review. I believe that the manuscript is greatly improved both in terms of the clarity of communication of the results and the reproducibility of the work.

I only have a couple of other minor comments.

1) It would be helpful to have a citation for MCMC model-data integration - despite being very familiar with many methods of Bayesian model inference and synthesis used in Epidemiology this particular flavour/terminology was new to me.

2) The reporting and discussion of the model fit could stand to be clearer. There is only a single data point for serology so it would be pretty difficult for a reasonable model not to be able to be calibrated to the observed value so it is not really relevant for assessing the goodness of fit of the model. With respect to the death trends "Closely reproduced reported death counts" (results section) is certainly not true for New York City and tonally quite different from "reasonably good fits" (discussion). For South Florida and Washington Puget Sound the observations are within 95% posterior predictive intervals (note not credible intervals as these are not model parameters), but for Washington arguably also fail to capture the qualitative shape of the death data. The fits are by no means bad or need to be improved (for the purposes here), but do warrant a more careful (and consistent) description and discussion particularly of the relatively poorer fit to NYC.

3) Figure 4: Top row, might help to clarify in legend that top row is plotted total cumulative deaths against the daily testing rate rather than against time (as with bottom row).

Response to Reviewer Comments

Reviewer #2 (Remarks to the Author):

The revised manuscript included results adding vaccination to model and the discussion about performance of antibody tests, it's great improved in my personal opinion.

In discussion section (Line 364), when authors mentioned 'antibody levels have been shown to wane after several months', 2 references should be cited for their description of antibody wane in early time.

1) Long QX, Tang XJ, Shi QL, Li Q, Deng HJ, Yuan J, Hu JL, Xu W, Zhang Y, Lv FJ, Su K, Zhang F, Gong J, Wu B, Liu XM, Li JJ, Qiu JF, Chen J, Huang AL. Clinical and immunological assessment of asymptomatic SARS-CoV-2 infections. *Nat Med*. 2020 Aug;26(8):1200-1204. doi: 10.1038/s41591-020-0965-6. Epub 2020 Jun 18. PMID: 32555424.

2) Ibarondo FJ, Fulcher JA, Goodman-Meza D, Elliott J, Hofmann C, Hausner MA, Ferbas KG, Tobin NH, Aldrovandi GM, Yang OO. Rapid Decay of Anti-SARS-CoV-2 Antibodies in Persons with Mild Covid-19. *N Engl J Med*. 2020 Sep 10;383(11):1085-1087. doi: 10.1056/NEJMc2025179. Epub 2020 Jul 21. Erratum in: *N Engl J Med*. 2020 Jul 23;: PMID: 32706954; PMCID: PMC7397184.

Response:

Thank you. These references have been added.

Reviewer #3 (Remarks to the Author):

First I would like to thank the authors for the comprehensive and detailed response to the questions and queries in my original review. I believe that the manuscript is greatly improved both in terms of the clarity of communication of the results and the reproducibility of the work.

I only have a couple of other minor comments.

1) It would be helpful to have a citation for MCMC model-data integration - despite being very familiar with many methods of Bayesian model inference and synthesis used in Epidemiology this particular flavour/terminology was new to me.

Response:

We have added the following references:

Ghassan Hamra, Richard MacLehose, David Richardson, Markov Chain Monte Carlo: an introduction for epidemiologists, *International Journal of Epidemiology*, Volume 42, Issue 2, April 2013, Pages 627–634, <https://doi.org/10.1093/ije/dyt043>

Philip D. O'Neill, A tutorial introduction to Bayesian inference for stochastic epidemic models using Markov chain Monte Carlo methods, *Mathematical Biosciences*, Volume 180, Issues 1–2, 2002, Pages 103-114, ISSN 0025-5564, [https://doi.org/10.1016/S0025-5564\(02\)00109-8](https://doi.org/10.1016/S0025-5564(02)00109-8).

2) The reporting and discussion of the model fit could stand to be clearer. There is only a single data point for serology so it would be pretty difficult for a reasonable model not to be able to be calibrated to the observed value so it is not really relevant for assessing the goodness of fit of

the model. With respect to the death trends "Closely reproduced reported death counts" (results section) is certainly not true for New York City and tonally quite different from "reasonably good fits" (discussion). For South Florida and Washington Puget Sound the observations are within 95% posterior predictive intervals (note not credible intervals as these are not model parameters), but for Washington arguably also fail to capture the qualitative shape of the death data. The fits are by no means bad or need to be improved (for the purposes here), but do warrant a more careful (and consistent) description and discussion particularly of the relatively poorer fit to NYC.

Response:

We agree that the reporting of these results was discrepant in different parts of the text and could be improved. We have replaced the section in the results with the following text which is more tonally consistent with the text in the Discussion describing fits.

"Model fits reproduced reported death trends reasonably well through June 2020 and seroprevalence estimates early in the outbreak. Of note, fits were poorer for New York City, which was not unexpected due to the unique severity of the initial pandemic wave there. Fits were moderately good for Washington Puget Sound and best for South Florida." (Results)

3) Figure 4: Top row, might help to clarify in legend that top row is plotted total cumulative deaths against the daily testing rate rather than against time (as with bottom row).

Response:

We have clarified in the figure legend.